# Dietary Quality and Perceived Barriers to Weight Loss among Older Overweight Veterans with Dysmobility

**DOI:** 10.3390/ijerph19159153

**Published:** 2022-07-27

**Authors:** Elizabeth A. Parker, William J. Perez, Brian Phipps, Alice S. Ryan, Steven J. Prior, Leslie Katzel, Monica C. Serra, Odessa Addison

**Affiliations:** 1Department of Physical Therapy and Rehabilitation Science, University of Maryland School of Medicine, Baltimore, MD 21201, USA; oaddison@som.umaryland.edu; 2Baltimore VA Medical Center GRECC, VA Maryland Health Care System, Baltimore, MD 21201, USA; william.perez5512@gmail.com (W.J.P.); bphipps28@gmail.com (B.P.); aryan@som.umaryland.edu (A.S.R.); sprior@umd.edu (S.J.P.); lkatzel@som.umaryland.edu (L.K.); 3Division of Gerontology, Geriatrics and Palliative Medicine, Department of Medicine, University of Maryland School of Medicine, Baltimore, MD 21201, USA; 4Department of Kinesiology, University of Maryland School of Public Health, College Park, MD 20742, USA; 5Division of Geriatrics, Gerontology & Palliative Medicine and the Sam & Ann Barshop Institute for Longevity & Aging Studies, UT Health San Antonio, San Antonio, TX 78229, USA; serram@uthscsa.edu; 6San Antonio GRECC, South Texas VA Health Care System, San Antonio, TX 78229, USA

**Keywords:** dietary quality, aging, veterans, impaired mobility

## Abstract

Healthier diets are associated with higher muscle mass and physical performance which may reduce the risk of developing frailty and disability later in life. This study examined the dietary quality and self-reported weight loss barriers among older (>60 years), overweight (BMI ≥ 25 kg/m^2^) Veterans with dysmobility (low gait speed, impaired mobility diagnosis, or a comorbidity that results in impaired mobility). Habitual dietary intake and healthy eating index (HEI-2015) were assessed using 24-h recalls and compared to US nationally representative dietary intake data and national recommendations. The “MOVE!11” Patient Questionnaire assessed weight loss barriers. The sample (*n* = 28) was primarily male (93%), black (54%) and obese (BMI = 35.5 ± 5.4 kg/m^2^) adults aged 69.5 ± 7.0 years with two or more comorbidities (82%); 82% were prescribed four or more medications. Daily intakes (mean ± SD) were calculated for total energy (2184 ± 645 kcals), protein (0.89 ± 0.3 g/kg), fruits (0.84 ± 0.94 cup·eq.), vegetables (1.30 ± 0.87 cup·eq.), and HEI-2015 (52.8 ± 13.4). Veterans consumed an average of 11% less protein than the recommendation for older adults (1.0 g/kg/d) and consumed fewer fruits and vegetables than comparisons to national averages (18% and 21%, respectively). Mean HEI-2015 was 17% below the national average for adults >65 years, suggesting poor dietary quality among our sample. Top weight loss barriers were not getting enough physical activity, eating too much and poor food choices. This data suggests that dietary quality is suboptimal in older, overweight Veterans with disability and highlights the need to identify strategies that improve the dietary intake quality of older Veterans who may benefit from obesity and disability management.

## 1. Introduction

Veterans are at high risk for obesity and related chronic diseases. Veterans treated within the U.S. Veterans Health Administration (VHA) are more likely to be overweight and have higher waist circumference than the general population [1,2]. This has also been demonstrated among older Veterans [3]. Veterans utilizing VHA facilities have more comorbid illnesses than individuals who receive outpatient care from non-VHA facilities [4].

Healthy diets are linked to reduced risk of obesity and chronic disease, yet little is known about the dietary habits of older, overweight Veterans. In general, older adults have unique dietary challenges due to a myriad of factors, including age-related sensory changes [5] and suboptimal nutrition knowledge [6], that increase the risk for poor dietary quality. The dietary intake of older adults tends to fall below the recommended amounts of total energy, fiber and micronutrients while exceeding recommended fat, sodium and added sugar [7]. The Healthy Eating Index (HEI) is a measure of adherence to the Dietary Guidelines that are commonly used to assess dietary quality. Higher HEI scores are associated with a risk reduction of all-cause mortality, and numerous chronic conditions [8,9]. Additionally, healthier dietary quality is associated with higher muscle mass, strength and physical performance [10,11,12], which may prevent or delay mobility limitation later in life [13]. On the contrary, lower dietary quality is associated with a higher risk for incident self-reported mobility limitations [14]. Poor mobility and multimorbidity also may adversely affect diet quality. Difficulties performing basic self-care tasks such as meal preparation or feeding oneself may negatively impact food and/or beverage consumption [15]. Veterans with dysmobility may be at particular risk for unhealthy diets contributing to overweight and obesity in this population.

Older Veterans have multiple unique obstacles to maintaining a healthy diet. In addition to established age-related factors that may impact dietary intake [5], the military environment experienced by Veterans emphasized maintaining optimal physical performance over nutrition education [16]. Mental health, education status, social support, socioeconomic status and food insecurity are also critical factors to consider [6,17,18]. In order to develop effective interventions to mitigate these factors, there must be an understanding of diet quality in older Veterans, particularly in those with limited mobility. Therefore, the purpose of this study was to examine diet quality and self-reported weight loss barriers in overweight, older Veterans with dysmobility enrolled in a weight loss study.

## 2. Materials and Methods

### 2.1. Study Design

This cross-sectional study combines baseline data from participants enrolled in one of two separate randomized controlled weight loss trials at a central location. Eligible participants for this report were Veterans > 60 years of age with impaired mobility, defined as having a low gait speed, a diagnosis of impaired mobility, or a comorbidity that results in impaired mobility who completed dietary recalls at baseline (*n* = 28). Study protocols were approved by the institutional review board of the University of Maryland School of Medicine and the Baltimore VHA. All participants provided written informed consent prior to study enrollment.

### 2.2. Measurements

Demographic data included sex, race/ethnicity and age at baseline visit. Medical charts were reviewed to gather chronic health conditions and VHA prescribed medications for each participant. Medications prescribed by non-VHA providers were included when available. Participants with two or more chronic health conditions were classified as having comorbidity. Polypharmacy was defined as a prescription for four or more medications; taking four or more medications has been associated with increased fall incidence and increased recurrent fall risk among older individuals [19].

Measures of height and weight were measured by a trained exercise physiologist with participants in a fasted state, wearing lightweight clothing and no shoes and used to calculate BMI (kg/m^2^).

Habitual dietary intake, including total daily energy intake (kcals), protein (g/day; g/kg body weight), fat (g/day; % energy), saturated fat (% energy), daily servings of total dairy, fruits and vegetables (cup equivalents), refined and whole grains (oz equivalents; % total grains from whole grains), added sugar (tsp) and sodium (mg) intakes, was assessed using an average of three, nonconsecutive 24-h recalls [20] from the Automated Self-Administered 24-h (ASA24) Dietary Assessment Tool (versions 2016 and 2018) developed by the National Cancer Institute, Bethesda, MD [21]. Participants completed each recall in the study facility assisted by a registered dietitian. Dietary recalls were used to calculate diet quality via HEI-2015 [22]. The HEI-2015 includes 13 components that are summed (range 0–100); higher total scores indicate better dietary quality. In an effort to provide a “rating” of the overall American diet, a grading scale was developed: >80 = “good”; 51–80 = “needs improvement”; <51 = “poor” [23].

Comparisons to national average intakes were made for available nutrient and food group equivalents using NHANES What We Eat in America data tables specifying intakes by gender and age [24]. The relative difference between our sample and from NHANES for each specified nutrient and food group equivalent was calculated ((Veteran intake − national average)/national average). We also calculated the percentage of Veterans who were consuming intakes above or below the NHANES average intake and average HEI score. Dietary intakes were compared to national recommendations by the US Dietary Guidelines [25] and the American Heart Association for age and gender (when applicable) [26]. Given concerns with age-related loss of muscle mass, we used optimal protein recommendations for older adults (1.0–1.3 g/kg body weight) as the reference for comparison to our sample [27]. To calculate the percentage of whole grain intake, we divided whole grain intake by total grain intake. Where there is a national recommendation available for comparison, we calculated the percentage of Veterans who were compliant with the recommendation.

To assess barriers that might negatively affect weight loss or making healthy dietary choices, a subset of this sample (57%; *N* = 16) answered questions on perceived barriers to weight loss from the “MOVE!11” Patient Questionnaire [28], and perceptions of weight management at the VHA adapted from Jay et al. [29]. The following questions measured food insecurity [30]: “*Within the past 12 months we worried whether our food would run out before we got money to buy more” and “Within the past 12 months the food we bought just didn’t last and we didn’t have money to get more*.” A response of “often true” or “sometimes true” to either question indicates a positive screen for food insecurity. The Newest Vital Sign screening tool assessed health literacy with the following scores defining literacy levels: ≤1, high likelihood of limited literacy; 2–3, possibility of limited literacy; ≥4, adequate literacy [31].

### 2.3. Statistical Analysis

Data were analyzed using SAS version 9.4 (SAS, Cary, NC, USA). Means and standard deviations were calculated for continuous variables and presented unless otherwise specified, and frequencies were calculated for categorical variables.

## 3. Results

### 3.1. Demographics

Overall, 28 participants completed three dietary recalls. Participant demographics are displayed in Table 1. The sample was primarily male (93%), black (54%) or white (46%) with a mean age of 69.5 ± 6.9 years. The mean BMI was 35.3 ± 5.1 kg/m^2^. Most of the sample had two or more comorbidities (82%), and 82% were prescribed four or more medications. All participants were community-dwelling individuals living independently at home.

### 3.2. Dietary Intake

The dietary intake results are presented in Table 2. Mean daily energy intake was 2183 ± 645 kcal, of which approximately one-third of total energy was from fat (36.4 ± 7.8%). Daily protein intake was 0.89 ± 0.30 g/kg body weight, and 68% consumed less than the recommended 1.0 g/kg/d of protein for older adults (data not shown). Veterans consumed fewer than three combined daily servings of fruits (0.8 ± 0.9 cup·eq.) and vegetables (1.3 ± 0.9 cup·eq.), and 96% consumed fewer than the recommended five daily servings of fruits and vegetables (data not shown). Veterans consumed a much higher proportion of total daily servings of grains from refined grains (5.4 ± 2.5 oz·eq.) versus whole grains (1.3 ± 1.2 oz·eq.) and consumed approximately 18 teaspoons of added sugar daily. Mean daily sodium intake was 4062 ± 1373 mg. For all nutrients and food group equivalents except fruits and vegetables, Veterans consumed mean intakes above the NHANES average. On average, Veterans consumed 18.6% more sodium and 17.9% more added sugars than the NHANES average but consumed fewer mean servings of fruits and vegetables (17.8% and 21.2%, respectively). The percentage of Veterans who consumed intakes above the NHANES averages ranged from 39% to 64% depending on the nutrient. The percentage of Veterans who consumed fewer servings of fruits and vegetables were 68% and 75%, respectively. Compared to national dietary recommendations, Veterans’ consumption exceeded the percentage of energy consumed from fat and saturated fat, and mean intakes of sodium and added sugars, whereas mean protein intake and the percentage of total grains consumed as whole grains were below national recommendations. The mean HEI score was 52.8 ± 13.4 (Range: 23.6–77.9). The majority of the sample (61%) had dietary intakes categorized as needing improvement, and 31% were categorized as having a poor dietary intake, reflecting a total of 92% who failed to have optimal HEI. The majority (79%) of our Veterans had an HEI score below 64, which was the national average for older adults.

### 3.3. Barriers to Weight Loss

Sixteen participants completed the questionnaires. The mean number of barriers reported from the “MOVE!11” survey was 3.1 ± 2.8 (range: 0–12). The most common barriers reported were not getting enough physical activity (69%), followed by eating too much (44%) and poor food choices or habits (38%) (Table 3). Slightly more than half (53%) of the sample was classified as having adequate literacy; 18% was classified as possibility of having limited literacy and 29% had a high likelihood of limited literacy. The majority of the sample reported food security: 71% responded ‘never true’ to the question, “Within the past 12 months we worried whether our food would run out before we got money to buy more”. None of our sample responded ‘often true’ to the question, “Within the past 12 months the food we bought just didn’t last and we didn’t have money to get more”; 21% responded ‘sometimes true’ to this statement.

## 4. Discussion

Overall, this study suggests that dietary quality is suboptimal in older, overweight Veterans with dysmobility. First, we found that on average, Veterans consumed 11% less protein than recommendations for older adults [27], and had 18% lower fruit and 21% lower vegetable intakes per day than national recommendations [25]. Additionally, the mean HEI score among this sample was 12 points (17%) below the national average for older adults [32], further indicating poorer dietary quality among our sample. These data are especially relevant in older Veterans, who are more commonly diagnosed as overweight or obese than the general population [3].

One possible reason for the increased obesity prevalence among older Veterans is the difficulty maintaining body weight following completion of service in a highly structured military environment emphasizing high levels of intense physical activity with less of a focus on nutrition education requirements [16]. Following completion of military service, physical activity is dramatically reduced, but a diet consisting of higher calorie, low nutrient-dense foods is maintained leading to energy imbalance and weight gain [33]. Prior reports demonstrate that while the majority of active-duty military personnel met the Healthy People 2010 objective for moderate/vigorous physical activity, the majority failed to meet the diet-related objectives for fruit, vegetable and whole-grain intake [34]. In response to these reports of poor dietary intake, the military began implementing initiatives to promote healthy eating behavior [35], but Veterans over the age of 60 years completed their service decades prior to the start of these initiatives.

In our sample, lack of physical activity was reported most often as a perceived reason for being overweight, but less than half attributed being overweight to poor food choices or habits. Similarly, a recent study found that Veterans consider physical activity to be the most important factor in weight loss and/or weight maintenance [29]. However, evidence suggests that nutrition and caloric intake has a larger impact on obesity reduction than physical activity [36]. Energy intake did not appear to be greater than recommendations; however, ~40% of Veterans reported that the cause of their excess body weight was due to eating too much, as well as poor food choices, suggesting that the quality of their diet (increased energy value) has likely influenced their weight gain. Other influences on diet from military experience may stem from traumatic moments during active duty. A study including Veterans from World War II to The Iraq War found that many Veterans developed hoarding eating habits from the eating environments in military settings [37]. Veterans were subject to developing insecurity-influenced eating behaviors from forced weight controls and wartime experiences such as being held in captivity (POW). Veterans describe their post-service readjustment as the time when addictions to food and substances impact their ability to maintain a healthy diet and weight [37]. It is likely that these habits are maintained through life although this is unknown. Although few in our sample reported that they had difficulty with self-control, approximately half reported that eating too much contributed to their overweight status. The culmination of the poor eating habits developed during military experiences and the combined changes to energy expenditure following completion of service create a scenario where Veterans are at a high propensity to become overweight and obese. These data indicate the need for Veteran education on the importance of nutrition in obesity management. This is especially relevant given that approximately half of our sample was classified as having limited or low literacy, which could further impact the Veteran’s ability to make or identify the link between healthier dietary choices and obesity status.

Age impacts numerous biological changes that involve the reduction of lean mass and redistribution of fat, reducing daily caloric expenditure and contributing to the rising prevalence of obesity among this age group [38,39,40]. Loss of lean mass may also be reflected in changes in dietary intake resulting in less optimal dietary quality. Drugs frequently prescribed to older adults (i.e., ACE inhibitors; calcium channel blockers; NSAIDs; steroids; anti-psychotics) can influence appetite and sense of smell [41]. In addition to the age-related biological factors affecting the diet quality of older adults, age-related socio-economic factors play a role. Many older adults modify eating habits based on changes in their financial and living situations [42]. Men living alone are more likely to have a poor diet than men living with a spouse, particularly those older than 74 years [43]. Our sample included community-dwelling older adults, so it is assumed that they are independently preparing their own meals however we cannot confirm whether this was truly the case. Future studies should include detailed assessments related to the ability to prepare meals in order to tailor interventions to account for these barriers. In addition, potential changes in income due to retirement or medical conditions that affect their ability to work can negatively influence the quality of food consumption [43]. Most participants in our study did not identify food insecurity as a problem, suggesting that income was not a factor when making food purchasing decisions. However, it is possible that our sample may not have identified with these questions (answering never true) as expected, given that their food choices are less expensive than healthy foods. Future studies among this population should explore how annual income impacts food purchasing behaviors. Qualitative data collected informally from this group of participants suggests that smoking cessation plays a role in food choices, particularly among those who were former long-term smokers (unpublished, Katz, 2020). The time spent smoking may have altered taste sensations, as participants subjectively report taste as a barrier to consuming healthier foods such as vegetables. Combined with age-related changes in taste and smell [42], these factors may negatively impact the quality and quantity of food consumption.

We acknowledge a few limitations in our study. Although there is a dearth of available data focused on the dietary quality of older Veterans with limited mobility, the cross-sectional analysis of dietary quality and reported barriers limits causality. In addition, the small sample size of this current study and urban location limits the generalizability to all older Veterans. However, the inclusion of older individuals with multiple comorbid conditions and polypharmacy is important given that these adults are often excluded from weight loss trials due to safety concerns with making dietary modifications and/or providing exercise prescriptions. A recent systematic review focused on behavioral weight loss interventions lasting 6 months or longer in older adults with obesity included six unique trials [44]. Of these, four trials excluded participants with diabetes and/or kidney disease. Other trials focused specifically on weight loss among type 2 diabetics include age restrictions [45], or studied populations with lower mean ages [46], limiting available research on adults >75 years [45]. There was a high prevalence of comorbidities including diabetes and kidney disease and polypharmacy among our older sample. Yet, these participants would greatly benefit from a dietary intervention. Physical activity was not assessed in this study as it was beyond the scope of these studies and thus, we cannot comment on their activity levels. However, physical activity is important for chronic disease prevention and should be included in future trials.

## 5. Conclusions

In conclusion, dietary intake quality is suboptimal in older, obese Veterans with disability. More specifically, Veterans in our study had lower dietary quality than the national average for older adults and did not achieve intakes of protein, fruits or vegetables based on national recommendations. The findings of this study may be beneficial when developing dietary interventions for older Veterans that address barriers such as current mental health status and lack of a support structure in order to adopt healthy dietary habits in addition to focusing on which dietary nutrients require modification. This study highlights the need to identify strategies that improve dietary quality with a specific focus on increasing dietary intake of protein, fruit and vegetables of older Veterans who may benefit from obesity and disability management. Effective strategies that target improvements in both physical activity and diet of older, overweight Veterans with dysmobility may be beneficial to combat the risk of obesity among this population.

## Figures and Tables

**Table 1 ijerph-19-09153-t001:** Demographics of older Veterans (*n* = 28) with dysmobility enrolled in an exercise and nutrition randomized controlled trial ^1^.

Sex, *N* (%)	
Male	26 (93)
Female	2 (7)
Race, *N* (%)	
Black	15 (54)
White	13 (46)
Age, years	69.5 ± 6.9
Body Weight, kg	106.2 ± 17.12
BMI, kg/m^2^	35.3 ± 5.1
Chronic Conditions, *N* (%)	
HTN	25 (89)
Hyperlipidemia	19 (68)
Diabetes	14 (50)
CKD	11 (39)
PAD	8 (29)
Comorbidity	23 (82)
Polypharmacy, *n* (%)	23 (82)

^1^ Mean ± SD unless otherwise indicated. BMI = body mass index, HTN = hypertension, CKD = chronic kidney disease, PAD= peripheral arterial disease, Comorbidity = at least 2 comorbid conditions, Polypharmacy = 4+ prescribed medications.

**Table 2 ijerph-19-09153-t002:** Mean dietary intake and HEI of older Veterans with dysmobility enrolled in an exercise and nutrition randomized controlled trial with comparisons to national averages and national dietary recommendations *.

	Nutrients and Food Group Equivalents	Older Veterans with Dysmobility	Comparison to Age-Matched Intake from NHANES [24]	Compliance (%) to National Recommendations of Older Veterans (*n* = 28)	% of Older Veterans with Intakes Above or Below Average NHANES Intake (*n* = 28)
Intakes above average NHANES intake	Energy, kcal	2184 ± 645	+6.0%	-	57%
	Protein, g	92.0 ± 28.5	+14.8%	-	64%
	Protein, g/kg BW 1.0–1.3 g/kg BW [27]	0.89 ± 0.30	-	36%	-
	Fat, g	89.3 ± 29.2	+6.8%	-	57%
	Fat, % energy 20–35% [25]	36.4 ± 7.8	-	36%	
	Saturated Fat, % energy <7% total kcals [26]	11.5 ± 3.2	-	4%	
	Total Dairy, cup eq.	1.5 ± 1.5	+10.5%	-	39%
	Refined Grains, oz eq.	5.4 ± 2.5	+3.1%	-	46%
	Whole Grains, oz eq.	1.3 ± 1.2	+2.5%	-	43%
	% total grains as whole grains >50% [25]	19.8 ±19.2	-	7%	-
	Sodium, mg <2300 mg [25]	4062 ± 1373	+18.6%	4%	57%
	Added sugars, tsp. <6 tsp females [26] <9 tsp males	17.8 ± 22.3	+17.9%	32%	43%
Intakes below average NHANES intake	Total Fruits, cup eq.	0.8 ± 0.9	−17.8%	-	68%
	Total Vegetables, cup eq.	1.3 ± 0.9	−21.2%	-	75%
	HEI Score and Components	Older Veterans with dysmobility	Comparison to age-matched intake from NHANES [32]	% of Older Veterans with HEI score < NHANES average	
	Total HEI score **	52.8 + 13.4	64.0	79%	
	*N* (%) total score, >80	0 (0)			
	*N* (%) total score, 51–80	17 (61)			
	*N* (%) total score, <51	11 (39)			

* Mean ± SD unless otherwise indicated. ** Maximum Possible Score = 100; higher scores indicate better adherence to the dietary guidelines.

**Table 3 ijerph-19-09153-t003:** Factors contributing to overweight status reported by older Veterans with dysmobility enrolled in an exercise and nutrition randomized controlled trial (*n* = 16).

Factor ^1^	% Reporting from “MOVE!11”	Why do Veterans Like Yourselves Tend to Be Overweight? ^2^
Not getting enough physical activity	68.7	“Lack of exercise” “Lack of exercise; can’t afford gym” “Inactivity” “Became a couch potato”
Eating too much	43.8	“Eating behavior same in AF academy”
Poor food choices or habits	37.5	
Boredom	31.3	
Love to eat	25.0	
Medications led to weight gain	25.0	
Eating because of emotions or stress	18.8	“Eat too much, stress”
Difficulty with self-control	12.5	“Careless, not focusing on it, giving into whims” “Food addict-myself, compulsive overeater from birth”
Feeling bad about myself	12.5	
Quitting tobacco use	12.5	
Loneliness or Loss of loved one	6.3	
Hungry all the time	6.3	
Illness or injury	6.3	“Disabling conditions” “Health issues that reduce movement” “Mobility/pain/depression”
Other	6.3	“Not regimented” “Losing regimented lifestyle after military service”
None of the above	6.3	

^1^ Participants were allowed to choose more than one response. ^2^ Open ended responses provided by Veterans answering the question, “Why do Veterans like yourselves tend to be overweight?”.

## Data Availability

Supporting deidentified data may be available to bona fide researchers upon request, subject to VA approval.

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
