# Peer review of "Dietary Quality and Perceived Barriers to Weight Loss among Older Overweight Veterans with Dysmobility"

_ijerph, 2022, doi:10.3390/ijerph19159153_

Round 1
Reviewer 1 Report
Greetings,
Your article presents valuable information. There are a few changes that need to be done:
Line 22- MOVE? what is it? I think it is an error. Please correct
Line 115- MOVE? what is it? I think it is an error. Please correct
Line 171- MOVE? what is it? I think it is an error. Please correct
Line 266- Conclusions on the next page
Line 311 si 312 - 1 appear before a name
Good luck!
Reviewer 2 Report
It is advisable to add information on whether the veterans were cared for. Did they live alone, with their family, in a nursing home, and did they have help preparing meals?
Conclusions should be more specific.
Reviewer 3 Report
The subject of the work seems to be purposeful, and the results, although obtained on a small number of people, may help in the development of dietary recommendations for older veterans with obesity and dysmobility.
Material and methods:
The chapter - Material and methods should include the scheme of the experiment and information about the number of people participating in each stage of the research and the method of their selection.
Why was the statistical analysis of the results not carried out, e.g. why was the relationship between the diet and BMI value, the severity of problems with mobility or comorbidity not compared?
Results:
For better analysis and discussion, the results should also be presented as a distribution of the obtained data, e.g. percentage of overweight, 1st and 2nd degree obesity, or percentage of people meeting and not meeting dietary recommendations, or people who eat less / more than the average intake in NHANES research, etc.
The description of Table 3 should contain information that it was possible to choose more than one answer.
With regard to the discussed importance of physical activity in increasing the risk of excessive body weight, it is worth referring to the level of physical activity of the participants in the study.
Discussion:
The authors should refer to the observations related to the energy intake (not different from the recommendations) and the respondents' declarations that the cause of their overweight / obesity is excessive food consumption, which means the increased energy value of the diet.
In the Discussion chapter, it is worth clarifying that the mean values of the obtained results are compared with the guidelines / other studies.
The statement: "Age impacts numerous biological changes that involve the reduction of lean mass and redistribution of fat, reducing daily caloric expenditure and contributing to the rising prevalence of obesity among this age group" should be referred to more recent publications.
Conclusions:
The conclusions should also contain information about support related to increasing the physical activity of older, overweight veterans with dysmobility.
